# Research on the influence mechanism of internet use on rural residents' consumption level in China——The mediating effect of consumption literacy

Zhen Tian[1☯], Rui Wang[2]*, Yan Tan[1☯]

1 Business School, Yangzhou University, Yangzhou, China, 2 Institute of Food and Strategic Reserves, Nanjing University of Finance and Economics, Nanjing, China

☯ These authors contributed equally to this work.

* wangfcb37@outlook.com

**Data Availability Statement:** All relevant data are within the paper and its Supporting Information files.

## Abstract

Based on the survey data of 936 rural households in Jiangsu Province, China in 2020, this paper empirically tests the influence mechanism of internet use on the rural residents' consumption level by constructing the theoretical analysis framework of "internet use—consumption literacy—rural residents consumption level" and taking consumption literacy as the mediator variable. The results demonstrate that, with other conditions unchanged, internet use has significantly improved the consumption level of rural residents. The mediating effect of the consumption literacy accounts for 15.08% among the entire effect of internet use on the rural residents' consumption level. Therefore, when applying digital information technology to improve the consumption level of rural residents in future, we should not only continue to increase investment in the construction of communication infrastructure in rural areas, but also continuously improve the consumption literacy of rural residents through building a multi-level complementary consumption education system and expanding the ways of consumption education and training.

## Introduction

As China's economy has entered into the stage of "New Normal" in recent years, how to stimulate consumption to form a new development pattern with domestic economic flow playing a leading role is crucial to achieve the high-quality and sustainable economic development [1]. However, since the Southeast Asian financial crisis in 1997, insufficient consumption, especially the low consumption level of rural residents, has been a key factor restricting China's economic growth [2]. By the end of 2019, China's rural population accounted for 40.28% of the total population, while the total retail volume to rural areas only accounted for 14.66% of the whole with the consumption level of rural residents being significantly on the lower side compared with that of urban residents. The main reasons are those like the relatively lower income level of rural residents, strong risk and uncertainty, unsound social security system,

**Funding:** This is a part research accomplishment of the project"Research on the Evolution of Rural Residents' Consumption Behavior and the Coordinated Policy of Supply and Demand in the Past Forty Years of Reform" (No. 18BJL004))", which is supported by Office of Chinese Philosophy and Social Sciences. The funder had a significant role in study design, data collection and analysis, decision to publish, or preparation of the manuscript.

**Competing interests:** The authors have declared that no competing interests exist.

and large income gap between urban and rural areas [3–6] There is another significant reason that the infrastructure conditions of rural transportation, logistics, communications are under-developed [7–9].

In order to further improve the consumption level of rural residents, since the release of the No.1 Central Document in January of 2008, the Chinese government has gradually shifted the focus of public infrastructure construction to rural areas, focusing on establishing a sound long-term investment guarantee mechanism, implementing village construction projects and promoting the upgrading of infrastructure construction such as rural waterways and power grids. In July of 2015, the Chinese government implemented the " Internet + " strategy nationwide and took the " Internet + three rural " as the focus of the entire action plan. In October, 2017, the report of the 19th National Congress of the Communist Party of China proposed to " promote the integrated development of the digital economy and the real economy with internet as the carrier and further enhance the basic role of consumption in the economic growth". In February, 2021, the No.1 Central Document further clarified the use of internet as a new engine to expand the consumption of rural residents and fully promoted the digital rural construction projects. With the promotion and wide use of internet in rural areas, all administrative villages have achieved connection to the broadband in December, 2021 with rural internet penetration rate reaching 59.2%, which further narrowed the urban-rural digital gap. With the improvement of internet application and the rapid development of rural e-commerce, the national rural network retail sales increased from 180 billion yuan in 2014 to 2050 billion yuan in 2021, with an average annual growth rate reaching 46.6%. Thanks to internet, the various economic and social resources have achieved interconnection, which effectively released the consumption potential of rural residents [10, 11]. It can be seen that, with the rapid development of the new economy symbolized by the wide application of information technology, internet is becoming a new engine to expand the consumption of rural residents [12, 13].

Researchers haven't reached theoretical consensus on the impact of internet use on the consumption level of rural residents and its impact mechanism. Some research found that internet use has significantly improved the consumption level of rural residents [14, 15]. However, there is also the view that the use of internet does not necessarily increase the total consumption level of residents because internet just plays the role in transferring traditional shopping channels to online, which is largely a substitute for offline consumption [16, 17]. The impact of internet use on the consumption level of rural residents is uncertain, and this uncertainty is closely related to the level of internet development [18]. Regarding the possible mechanism of the effects of internet use on the rural residents' consumption level, some research found that internet use increases the marginal productivity of agricultural production and the probability of non-agricultural employment and promotes the diversified household income sources of rural residents through improving both human and social capital of rural residents. This kind of employment effect and income effect brought by internet use have significantly led to the increase of the consumption level of rural residents [19–22]. Other research stated that internet use improves consumption efficiency, releases consumption potential and increases the consumption level of rural residents by reducing the information asymmetry, cutting the transaction cost and lessening the liquidity constraints [23–25].

The rapid development of the new economy has disrupted the traditional economic structure and the concept of new consumption driven by digital information technology has also appeared. New consumption requires residents to have better consumption literacy because it doesn't simply mean the replacement of traditional offline consumption with online consumption, which actually includes the innovation in consumption areas and consumption patterns [26, 27]. Residents' consumption literacy determines their consumption area and

consumption mode, thus affecting their consumption level [28, 29]. Recently, under the circumstances that internet is naturally technology-driven and the increasing technology level of consumption products proposes higher requirements on consumers, some academic researchers began to study the effects and influences of residents' consumption literacy when conducting research on the effects of internet on residents' consumption level. However, the existing research still lacks the mechanism analysis and the related discussion of internet use affecting the consumption level of rural residents by affecting their consumption literacy. As one of the relatively developed provinces in eastern China, Jiangsu has reached 76.7% internet penetration rate in 2020, ranking the fourth in China's internet development index and the first in the rural user amount with fixed internet broadband access in the country. In 2019, Jiangsu 's rural network retail sales accounted for 15.1% of the whole country 's amount. Furthermore, the uneven level of economic development in Jiangsu can be seen as a microcosm of China's economic development, which can be divided into northern, central, and southern regions according to the degree of economic development from low to high. Therefore, the manuscript uses questionnaire survey data from Jiangsu Province to empirically study the influence mechanism of internet use on rural residents' consumption level, which has certain rationality, and the research conclusions obtained based on this also have certain credibility.

The contributions of this paper mainly include the following aspects. (1) Based on the concept of consumer literacy, the research group designed 20 specific questions for measuring consumer literacy from five dimensions: consumer attitude, consumer motivation, consumer knowledge, consumer skills, and consumer traits. Drawing on the calculation method of financial literacy, the consumer literacy index was constructed to comprehensively evaluate the level of residents' consumer literacy, thus enriching the theory of consumption economy. (2) Based on the survey data of rural households in Jiangsu Province, the manuscript empirically tested the influence mechanism of internet use on the rural residents' consumption level by constructing the theoretical analysis framework of "internet use—consumption literacy—rural residents consumption level" and taking consumption literacy as the mediator, thus expanding academic research on rural household consumption from a micro perspective. (3) By studying the influence mechanism of internet use on the consumption level of rural households and formulating relevant countermeasures and suggestions based on research conclusions, it not only helps to stimulate the consumption potential of rural residents, but also has extremely important practical significance and policy value for building a new development pattern centered on expanding the domestic circulation and achieving high-quality economic development in China.

## Theoretical analysis and research hypothesis

Improving the consumption level of rural residents is the premise and foundation for forming China's domestic circulation and realizing high-quality and sustainable economic development. Internet use has changed the business environment and competition mode of enterprises, enabling them to provide products that truly meet the consumption needs of rural residents at a lower price. Meanwhile, it also breaks various restrictions that hinder consumption, effectively transforming the consumption needs of rural residents into real consumption behavior and improving the consumption level of rural residents. In addition, internet is a powerful tool for rural residents to learn consumption knowledge and skills and improve consumption attitudes and motives, which has a significant role in promoting the improvement of their consumption literacy and enables rural residents to improve consumption level in the course of pursuing higher-level consumption satisfaction.

## (1) Internet use and rural residents ' consumption level

According to western classical consumption theories, the main factors affecting the consumption level of a country's residents consist of income, wealth, consumption habits, psychological expectations, uncertainty and interest rates [30–34]. In addition, the impact of technological innovation cannot be ignored [35]. Every big improvement in consumption level in history can ultimately be attributed to technological innovation. Technological innovation has a significant positive impact on the long-term stable growth of residents' consumption levels [36, 37]. As the greatest invention of mankind in the 20th century, internet has provided inclusive information resource access facilities. Its promotion and application have initiated a new round of technological innovation, changed the business environment and competition way, promoted the development of economic globalization, and brought about tremendous changes in production and lifestyle. Especially after the ' Internet + ' was taken as the national strategy in 2015, there have been fundamental changes in product development, production process and marketing promotion, which in turn impacts the consumption level of rural residents from the supply side. Firstly, enterprises can use internet to automatically collect, retain and analyze the rural residents' information of consumption habits and preferences, promote the goods that match the consumption needs of rural residents in time, and make precise predications and prospective R&D on rural residents' consumption goods so as to cultivate and develop brand-new rural consumer favorites [38].Secondly, thanks to the internet, enterprises can build a platform for communication and interaction with rural residents, invite rural residents to participate in R&D design and production process of products they intend to buy [39], and realize the "personalized" private customization [40]. Thirdly, enterprises can use the internet to reduce the cost of production, management and marketing and then reduce the market price of goods suitable for rural residents consumption so as to relax the consumption budget constraints of rural residents and release more rural residents ' consumption potential [41].

For rural residents, internet has been embedded in all aspects of their consumption life, directly affecting their consumption level. Firstly, the bilateral trading platform on internet provides rural residents with richer and more comprehensive commodity information than offline physical consumption means, which is conducive to rural residents to quickly and easily select the products they need; Secondly, internet is a big supplement to the rural commercial construction, expanding the effective supply channels of consumption goods, promoting the sinking of consumption channels and breaking the restrictions on geographical and time conditions that hinder the consumption of rural residents; Thirdly, internet promotes the integration of online and offline consumption channels [42], which can help rural residents quickly and comprehensively collect the information regarding detailed usage and experience sharing of related products, thus reducing information gap. Meanwhile, it supports direct transactions with merchants through online payment, which greatly reduces the cost of transportation cost, time cost and transaction cost of rural residents' consumption and effectively transforms consumption needs into real consumer behavior, thus increasing rural residents ' consumption.

Based on this, this paper proposes the following Hypothesis 1:

H1: Internet use significantly increases the consumption level of rural residents.

## (2) Internet use, consumption literacy and consumption level of rural residents

Consumption literacy is a collection of knowledge, skills, attitudes, motivations and other related qualities that affect the consumption ability of consumers in the process of generating and meeting their high-level consumption needs [43]. In Marx's earliest study of consumption,

he pointed out that "Consumption is not abstinence, but the development of productive forces, that is, the ability to develop production. So it is both the resource and the ability to develop consumption". Theoretically speaking, consumption ability includes purchasing power, which is consumer's ability to pay and the premise of consumption. However, Marx believes that consumption ability is the embodiment and development of personal ability, which is actually the consumption quality of consumers. In order to increase residents' consumption level, it is more important to improve residents "consumption literacy" and make them become "comprehensive and free" people in addition to developing productivity to enrich consumption materials [44]. Nowadays, high technology is increasingly penetrating into the field of consumption and continuous improvement of technological content of consumption goods puts forward higher requirements for consumers. Rural residents with higher consumption literacy not only have more, newer and higher consumption needs for consumption goods, but also have more complete knowledge and talents on how to use consumption goods so as to fully utilize the use value of consumption goods. Therefore, in the context of the rapid development of new economy, the improvement of consumption literacy can organically integrate the development of rural residents themselves with their consumption behavior and provide a sustainable and strong impetus for the improvement of consumption level, which is the necessary condition and primary means for the improvement of consumption level. At the same time, as China's rural areas in the 21st century move from a traditional acquaintance society to a semi-acquaintance society [45], the use of internet has promoted the instant dissemination, free flow and universal sharing of information, knowledge and technology [46, 47]. Meanwhile, it has realized the upgrading of rural social communication levels from towering to flattening, reshaping the dimensions of interaction between rural residents and society [48]. This change in the communication mode of rural residents driven by internet use not only broadens the channels for rural residents to obtain information, but also expands the means and methods for them to learn various new knowledge and skills. Internet use is increasingly becoming a powerful tool for improving rural residents' consumption knowledge and consumption skills [49].

In addition, the transaction environment of internet use itself guides rural residents to complete the whole process covering information search, commodity selection, online payment and after-sales service during the process of consumption. These consumption experiences driven by internet technology will gradually transform into rural residents' knowledge reserve, skill improvement and experience accumulation, thus improving the consumption literacy of rural residents to a certain extent. In addition, the awareness effect and recommendation effect of internet advertising promotion, self-media marketing and online social sharing make it possible for open sharing of consumption knowledge and experience to spread unlimitedly through internet, which can bring rural residents the sense of pleasure, delicacy and sophistication brought by consumption, gradually embed new consumption concepts into the daily life of rural residents, change their consumption attitudes and consumption motives, and promote their consumption literacy to continuously pursue higher levels of consumption needs on the basis of availability of sufficient food and clothing [50]. It can be seen that internet use has effectively improved the consumption literacy of rural residents, thus further improving the consumption level of rural residents.

In summary, this paper proposes the following Hypothesis 2:

H2: Internet use can improve the consumption literacy of rural residents, and then affect the consumption level of rural residents. Consumption literacy plays an intermediary role in the impact of internet use on the consumption level of rural residents.

## Data source, model setting and variable selection

### (1) Data sources

Part of the data used in this paper comes from the questionnaire survey on rural residents' internet use and consumption conducted in Jiangsu Province by the researchers in July, 2020. The survey used a multi-stage stratified random sampling method. The first step was to randomly select three cities from different regions of Jiangsu Province, namely Suqian City in northern Jiangsu, Yangzhou City in central Jiangsu and Wuxi City in southern Jiangsu. In the second step, according to the statistics of per capita GDP in 2019, the counties (districts) included in the sample cities were divided into three groups which were high, medium and low. Three townships were randomly selected in each group. In the third step, the same method was used to sample 10–15 rural families randomly to make face-to-face structured interviews from the randomly sampled administrative villages. Respondents of this survey are the head of each household. The survey mainly includes the basic situation of rural residents and their families, the use of internet, the status of consumption literacy and various household consumption expenditures. A total of 957 rural households were interviewed and 936 valid questionnaires were obtained.

### (2) Model setting and variable selection

In order to empirically examine the impact of internet use on the consumption level of rural residents, this study mainly constructs two models. The first model is the core benchmark model, which is mainly used to identify the direct impact of internet use on the consumption level of rural residents. The setting is shown in Eq (1):

$$RRC_i = \alpha_0 + \alpha_1 IU_i + \alpha_2 CV_i + \varepsilon_{1i} \tag{1}$$

Secondly, in order to test the mechanism of internet use affecting the consumption level of rural residents by improving their consumption literacy, the following model is further constructed based on the mediating effect model [51].

$$RRCL_i = \lambda_0 + \lambda_1 IU_i + \lambda_2 CV_i + \varepsilon_{2i} \tag{2}$$

$$RRC_i = \eta_0 + \eta_1 IU_i + \eta_2 RRCL_i + \eta_3 CV_i + \varepsilon_{3i} \tag{3}$$

In the above model, $RRCi$ is the explained variable expressed with the per capita actual consumption expenditure of rural households, representing the consumption level of the $i$th rural residents. $IUi$ is taken as the core explanatory variable, representing the degree of internet use of the $i$th rural residents. This variable is measured by the number of years when the head of the household actually uses the internet. The longer the number of years of internet use is, the deeper the degree of internet use is. $CV_i$ is regarded as the control variable vector. Referring to the existing relevant literature, this paper selects three types of control variables including personal characteristics, family characteristics and village characteristics. The individual characteristics mainly include the gender, age, square of age, education level, marital status and health status of the head of household. Regarding family characteristics, the basic situation of rural households is represented by the total family population, the proportion of labor force, the proportion of children's dependency ratio, the proportion of elderly dependency ratio and the proportion of healthy members. The social capital status of rural households is measured by whether the surname is the most important of all in the village and whether there are members from formal organizations in the family. The proportion of the family members with higher education background is used to measure the human capital of the family. The per capita income, deposits and total cash of the family are used to measure the income and wealth of

the family. The village characteristics are measured by the distance between the village and the town. $\alpha_0$, $\alpha_1$, $\alpha_2$, $\lambda_0$, $\lambda_1$, $\lambda_2$, $\eta_0$, $\eta_1$, $\eta_2$ and $\eta_3$ are the estimated coefficients of the model. $\varepsilon_{1i}$, $\varepsilon_{2i}$ and $\varepsilon_{3i}$ are the random error terms of the model. $RRCL_i$, as a mediator variable, represents the consumption literacy of the $i$th rural residents. The specific calculation formula of consumer literacy index is developed as follows through applying Sarma's measurement method [51].

$$RRCL_i = 1 - \frac{\sqrt{(1-x_{1i})^2 + (1-x_{2i})^2 + (1-x_{3i})^2 + \cdots\cdots + (1-x_{mi})^2}}{\sqrt{m}} \qquad (4)$$

Quorum $x_{mi}$ denotes the score of the $i$th rural resident on the $m$th question (answer 'yes' got 1 point, and answer 'no' got 0 point). $m$ represents the total number of problems. According to the previous research, consumption literacy includes consumption knowledge, consumption skills, consumption attitude, consumption motivation and other related characteristics that may affect consumption ability. Consumption knowledge refers to consumer 's understanding and mastery of some basic consumption knowledge in the consumption process. Referring to the existing research findings, this paper mainly designed the following four questions for measurement: "Do you have a specific household consumption expenditure plan?", "Are you familiar with some common e-commerce platforms, such as Taobao, Jingdong and Pinduoduo?", "Do you know bank's policy on consumption credit?", "When buying goods, will you care about the information like shelf life and safety labeling?". Consumption skills refer to the skills of product selection, purchase, use and after-sales in the consumption process. Referring to the existing research findings, this paper mainly designed the following four questions for measurement: "When doing shopping, will you be able to buy goods according to the original plan rather than buy impulsively?", "When doing shopping, can you accurately collect relevant information and make decisions?", "Can you skillfully use non-cash payment methods such as WeChat and Alipay?", "Do you know how to ask for coordination and help when there is a problem with the purchased goods?". Consumption attitude refers to the consumer's cognitive evaluation, emotional feeling and action tendency of consumption behavior. The research group mainly designed the following four questions for measurement with reference to the existing research results: "If you have spare money, are you willing to use it for consumption to improve the quality of life rather than save them all?", "When you see some goods you like, will you buy it immediately if the price is appropriate?", "Does your family always have clear consumption goals and consumption priorities in different periods?", "Do you think credit consumption (e.g., mortgage, car loan, credit card consumption, etc.) is a good way of consumption?". The definition, assignment and descriptive statistics of each variable are shown in Table 1.

## (3) Endogenous problem discussion and instrumental variable selection

According to the previous analysis, internet use affects the consumption level of rural residents, and rural residents with high consumption level and good economic conditions are more likely to purchase computers, mobile phones and other electronic devices to make full use of internet. This two-way causality will lead to potential endogenous problems in the above model. In addition, there are many factors affecting the consumption level of rural residents, some of which may also affect the use of internet. In order to avoid the estimation bias caused by such problems, this paper applies the practice of Dettling(2017), Zhang(2019) [52, 53] by removing communication cost from the household consumption expenditure of rural residents, using the average years of internet use of other rural residents in the village (community) where the sample rural residents are located as instrumental variables to deal with endogenous problems so as to obtain a consistent estimation of parameters.

**Table 1. Variable definition, assignment and descriptive statistical analysis.**

| Variable | Item | Variable definition and assignment | Mean value | Standard deviation |
|---|---|---|---|---|
| Explained variable | Rural residents' consumption level | Household per capita consumption expenditure (logarithm) | 9.909 | 0.721 |
| Explanatory variables | Internet use | Years of Internet use (years) | 7.058 | 5.520 |
| Mediator variable | Consumer literacy | Calculated according to Formula (4) | 0.400 | 0.191 |
| Personal characteristics | Sexuality | 1 = male, 0 = female | 0.565 | 0.496 |
| | Age | | 51.258 | 14.982 |
| | The square of age | The square of the actual age | 2627.383 | 224.460 |
| | Level of education | Number of school years (years) | 9.302 | 3.915 |
| | Marital status | 1 = married, 0 = unmarried | 0.859 | 0.348 |
| | Health conditions | 1 = healthy, 0 = unhealthy | 0.894 | 0.308 |
| Family characteristics | Number of family members | Total number of family members (persons) | 3.469 | 1.395 |
| | Proportion of family labor force | The number of family labor (16 < x < 64) / family members (%) | 0.692 | 0.323 |
| | Children's dependency ratio | Children (≤ 16) / family labor force (%) | 0.164 | 0.273 |
| | Elderly dependency ratio | Number of elderly (≥ 65) / number of family labor force (%) | 0.148 | 0.318 |
| | Proportion of healthy members | Number of healthy members / total family population (%) | 0.902 | 0.239 |
| | Is the surname a village surname | 1 = yes, 0 = no | 0.538 | 0.499 |
| | Are there any organized members in the family | 1 = yes, 0 = no | 0.285 | 0.452 |
| | The proportion of higher education | Number of people with higher education / Total family population (%) | 0.182 | 0.235 |
| | Family average income | Total household income / Total household population (logarithm) | 10.399 | 1.228 |
| | Total deposits and cash | 1 = 50 000 and below, 2 = 50 ~ 100 000, 3 = 100 ~ 200 000 4 = 200,000–500,000,3 = more than 500,000 | 2.614 | 1.238 |
| Village characteristics | The distance between village and town | Lining material | 4.688 | 5.913 |

The main reasons for using this instrumental variable are as follows. On the one hand, from the perspective of correlation, rural residents are easily affected by others in the same village (community) in terms of the internet use due to the similarity of network infrastructure environments in the same village (community) as well as the function of "peer effect". The deeper the internet use of other rural residents in the village (community) is, the more possible it is for this rural resident to make response. The internet use of other rural residents in the same village (community) will not directly determine the consumption level of the rural residents, which satisfies the correlation and exclusiveness. On the other hand, from the exogenous point of view, the average number of years that other rural residents in the same village (community) use internet will not directly affect the consumption level of the rural residents, which is also not related to the unobservable factors that affect their consumption level. So it is a strict exogenous variable.

## Empirical results and analysis

### (1) The impact of internet use on the consumption level of rural residents

In this paper, Stata16.0 is used to make regression analysis of Model (1), and the results are shown in Table 2. OLS regression is used in the Regression (6). It can be seen from the results that the impact of internet use on the consumption level of rural residents is significant at the statistical level of 1% and the coefficient is positive, which indicates that the deeper the level of a rural resident's internet use is, the higher consumption level this resident has. Regarding the marginal effect, the consumption level of rural residents will increase by 1.9% for each

**Table 2. Regression estimation results of Internet use on rural residents' consumption level.**

| | Internet use | Rural residents' consumption level | | | | |
|---|---|---|---|---|---|---|
| | IV-2SLS First stage regression (1) | IV-2SLS (2) | IV-2SLS (3) | IV-2SLS (4) | IV-2SLS (5) | OLS (6) |
| Internet use | — | 0.054*** (0.013) | 0.042** (0.019) | 0.037** (0.018) | 0.036** (0.017) | 0.019*** (0.005) |
| The average number of years of internet use by other rural residents in the village (community) | 0.382*** (0.056) | — | — | — | — | — |
| Personal characteristics variables | Control | — | Control | Control | Control | Control |
| Family characteristics variables | Control | — | — | Control | Control | Control |
| Village characteristics variables | Control | — | — | — | Control | Control |
| Constant term | Control | Control | Control | Control | Control | Control |
| Kleibergen-Paap Wald rk F | — | 98.210 | 51.977 | 48.823 | 47.473 | — |
| Stock-Yogo10% Horizontal bias value | — | 16.38 | 16.38 | 16.38 | 16.38 | — |
| $R^2$ | 0.436 | 0.092 | 0.166 | 0.286 | 0.296 | 0.305 |
| Number of samples | 936 | 936 | 936 | 936 | 936 | 936 |

Note

\* \* \*, \* \* and \* are statistically significant at 1%, 5% and 10% respectively. The values in parentheses are robust standard errors.

additional year of internet use. In order to avoid the estimation bias caused by endogenous problems, the instrumental variables are further used to make two-stage least squares (2SLS) estimation. In Regression (1) achieved in the first stage of 2SLS analysis, there is a significant positive correlation between the average number of years of internet use of other rural residents in the village (community) and that of sample rural residents, which satisfies the correlation hypothesis of instrumental variables. With control variables added layer by layer in the second stage of 2SLS regression analysis, the results show that internet use has a robust positive impact on the consumption level of rural residents in terms of impact direction and significance. At the same time, the RKF test statistics estimated by IV-2SLS pass the test. Therefore, after the possible estimation errors caused by the endogenous problem are corrected, the coefficient of the explanatory variable of internet use changes from 0.019 to 0.036, which is significantly positive at the 5% level. The research results are relatively robust, which verify the above research hypothesis H1.

## (2) The mediating effect test of consumption literacy

Among Table 3 of the estimation result of Eq 2, the instrumental variable is the average number of years that other rural residents in the village (community) has used internet, which is tested to meet the correlation hypothesis without weak instrumental variable problem. After the estimation errors that may be caused by the endogenous problem are corrected, the coefficient of the explanatory variable of internet use changes from 0.010 to 0.009, which indicates that there is overestimation on the causal processing effect in the OLS estimation. However, the degree of overestimation is small with no serious endogeneity. The research results are relatively robust with internet use having a positive impact on rural residents' consumption literacy at a significant level of 5%.

Among Table 4 about the regression results of Model (3), the instrumental variable is the average number of years of internet use by other rural residents in the village (community). According to the estimation results in Column (5), internet use and consumption literacy both have significant positive impacts on the consumption level of rural residents with estimated coefficients of 0.033 and 0.355 respectively.

**Table 3. Regression results of internet use on rural residents' consumption literacy.**

| | Internet use | Consumption literacy | | | | |
|---|---|---|---|---|---|---|
| | IV-2SLS First stage regression (1) | IV-2SLS (2) | IV-2SLS (3) | IV-2SLS (4) | IV-2SLS (5) | OLS (6) |
| Internet use | — | 0.019*** (0.003) | 0.009** (0.005) | 0.009** (0.004) | 0.009** (0.004) | 0.010*** (0.001) |
| The average number of years of Internet use by other rural residents in the village | 0.382*** (0.056) | — | — | — | — | — |
| Personal characteristics variables | Control | — | Control | Control | Control | Control |
| Family characteristics variables | Control | — | — | Control | Control | Control |
| Village characteristics variables | Control | — | — | — | Control | Control |
| Constant term | Control | Control | Control | Control | Control | Control |
| Kleibergen-Paap Wald rk F | — | 98.210 | 51.977 | 48.823 | 47.473 | — |
| Stock-Yogo 10% Horizontal bias value | — | 16.38 | 16.38 | 16.38 | 16.38 | — |
| $R^2$ | 0.436 | 0.310 | 0.407 | 0.437 | 0.438 | 0.438 |
| Number of samples | 936 | 936 | 936 | 936 | 936 | 936 |

Note

\* \* \*, \* \* and \* are statistically significant at 1%, 5% and 10% respectively. The values in parentheses are robust standard errors.

After completing step-by-step test, we make the confidence interval test as a substitute test and apply the coefficient product method to calculate the proportion of mediation effects through referring to the mediation effect test steps of Wen Zhonglin (2014) [46]. The results are as shown in Table 5. First of all, it can be seen from Column (1) of Table 5 that the coefficient of internet use in Model (1) is significantly positive at the level of 10%, which is based on the mediating effect. Secondly, the regression coefficients of Model (2) and Model (3) corresponding to Column (2) and Column (3) in Table 5 are both significant, which indicates that the indirect effect is significant. Thirdly, the coefficient of internet use, 0.030, is significant at

**Table 4. Regression results of the impact of internet use and consumption literacy on the consumption level of rural residents.**

| | Internet use | Rural residents' consumption level | | | | |
|---|---|---|---|---|---|---|
| | IV-2SLS First stage regression (1) | IV-2SLS (2) | IV-2SLS (3) | IV-2SLS (4) | IV-2SLS (5) | OLS (6) |
| Internet use | — | 0.043** (0.018) | 0.036* (0.020) | 0.033* (0.019) | 0.033* (0.019) | 0.014*** (0.005) |
| Consumer literacy | — | 0.604** (0.303) | 0.603*** (0.216) | 0.343* (0.196) | 0.355* (0.195) | 0.501*** (0.126) |
| The average number of years of Internet use by other rural residents in the village | 0.382*** (0.056) | — | — | — | — | — |
| Personal characteristics variables | Control | — | Control | Control | Control | Control |
| Family characteristics variables | Control | — | — | Control | Control | Control |
| Village characteristics variables | Control | — | — | — | Control | Control |
| Constant term | Control | Control | Control | Control | Control | Control |
| Kleibergen-Paap Wald rk F | — | 51.226 | 42.952 | 41.414 | 39.989 | — |
| Stock-Yogo10% Horizontal bias value | — | 16.38 | 16.38 | 16.38 | 16.38 | — |
| $R^2$ | 0.436 | 0.127 | 0.187 | 0.294 | 0.305 | 0.315 |
| Number of samples | 936 | 936 | 936 | 936 | 936 | 936 |

Note

\* \* \*, \* \* and \* are statistically significant at 1%, 5% and 10% respectively. The values in parentheses are robust standard errors.

**Table 5. The mediating effect of consumption literacy in the impact of internet use on the consumption level of rural residents.**

| | Rural residents' Consumption level | Consumption literacy | Rural residents' Consumption level |
|---|---|---|---|
| | IV-2SLS (1) | IV-2SLS (2) | IV-2SLS (3) |
| Internet use | 0.036* (0.019) | 0.009** (0.005) | 0.030* (0.018) |
| Consumer literacy | — | — | 0.603*** (0.128) |
| Mesomeric effect ($\lambda_1\eta_2$) Significance and confidence interval | 0.006* (p-value = 0.066) [0.001,0.011] | | |
| Mesomeric effect (%) | 15.075 | | |
| Other control variables | Control | Control | Control |
| Constant term | Control | Control | Control |
| $R^2$ | 0.295 | 0.392 | 0.311 |
| Number of samples | 936 | 936 | 936 |

Note

$*\,*\,*$, $*\,*$ and $*$ are statistically significant at 1%, 5% and 10% respectively. The values in parentheses are robust standard errors.

the level of 10% in the regression results of Model (3) listed in Table 5, which indicates that the direct effect is also significant. $\lambda_1\eta_2$ With further calculation, the result is 0.005, which is the same as the coefficient used for internet use in Model (3). As they both are positive, it is considered to have played partial mediating effect. Finally, the test result of $\lambda_1\eta_2$ is also significantly positive at 5% level. By applying the Bootstrap method to self-sample 1000 times so as to directly test the significance of the product of coefficients, the result demonstrates that the confidence interval is [0.001,0.011], which proves that the mediating effect still exists. According to the coefficient product method, the mediating effect of consumption literacy accounts for 15.08% in the total effect of internet use on the consumption level of rural households. Therefore, internet use affects the consumption level of rural residents by affecting the consumption literacy of rural residents. About 15.08% of the impact of internet use on the consumption level of rural residents is realized through the intermediary role of consumption literacy, which verifies the research hypothesis H2.

## (3) Heterogeneity analysis

From the perspective of macro level of Jiangsu Province, internet use can effectively promote the consumption level of rural residents. However, due to the uneven economic development in various regions of Jiangsu Province, the use of sub-samples in different regions for testing may lead to the results of regional heterogeneity test. It can be seen from Table 6 that the

**Table 6. Regression results of regional heterogeneity test.**

| | Rural residents' consumption level | | |
|---|---|---|---|
| | Southern Jiangsu | Central Jiangsu | Northern Jiangsu |
| Internet use | 0.0416*** (0.012) | 0.029*** (0.009) | 0.0261*** (0.006) |
| Personal characteristic variables | Control | Control | Control |
| Family characteristic variables | Control | Control | Control |
| Village characteristic variables | Control | Control | Control |
| Constant term | Control | Control | Control |
| $R^2$ | 0.342 | 0.251 | 0.276 |
| Number of samples | 342 | 304 | 290 |

Note

$*\,*\,*$, $*\,*$ and $*$ are statistically significant at 1%, 5% and 10% respectively. The values in parentheses are robust standard errors.

estimated coefficient of internet use is significantly positive at the level of 1%, which demonstrates that internet use is not influenced by the regional difference in the current rapid development of the digital economy and imposes certain impact on the overall rural residents' consumption in the province.

## Conclusions and recommendations

Based on the questionnaire survey data of 936 rural households in Jiangsu Province, this paper empirically studies the function mechanism of the internet use on the consumption level of rural residents and applies the instrumental variable method to solve the possible estimation bias caused by endogenous problems. The results demonstrate the following findings. Firstly, under the premise of other conditions unchanged, internet use is conducive to promoting the improvement of rural residents' consumption level; Secondly, internet use has improved the consumption literacy of rural residents, thereby improving the consumption level of rural residents. Among the total effects of internet use on the consumption level of rural residents, the mediating effect of consumption literacy accounts for 15.08%.

Based on the above conclusions, the following policy measures are suggested to effectively improve the consumption level of rural residents in the context of the rapid development of the internet. Firstly, more investment should be made in communication infrastructure construction in rural areas so as to improve the service system of mobile devices and household broadband, improve the quality and level of internet operations and services, reduce the threshold of internet use, and solve the practical problems of rural residents such as "being difficult to access internet", "being expensive to access internet", and "having slow access to internet"; Secondly, considering the features of internet consumption patterns and all segments of internet consumption, the government should formulate and improve the relevant laws and regulations, strengthen the supervision on all segments of internet consumption, improve the consumption supervision system and regulate the internet commodity market order so as to provide a good internet consumption environment for rural residents; Thirdly, on the purpose of improving the rural residents ability to understand and use internet comprehensively and helping them adapt to and integrate themselves into the digital life gradually, internet application training should be carried out regularly or irregularly in the community center or village committee through various means like the popularizing internet knowledge and making practical operation and simulation; Fourthly, the ways of consumer education and training should be expanded through building a complementary multi-level consumer education system covering schools, families and society so as to enrich the knowledge and skills of rural residents in terms of the identification, analysis, evaluation, rights protection and full application of commodity; Fifthly, the media's role in guiding public opinions should be effectively strengthened to help rural residents fully realize that the essence of consumption is to make people become "comprehensive and free" so as to cultivate their scientific and rational consumption concept and attitude and encourage them to continuously improve their consumption level and quality of life by satisfying more, newer and higher consumption needs.

## Supporting information

**S1 Data.**
(DTA)

## Author Contributions

**Data curation:** Rui Wang.

**Writing – original draft:** Zhen Tian.

**Writing – review & editing:** Yan Tan.

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
