## [Decision Letter · Decision Letter 0]

15 Aug 2023

PONE-D-23-22564Research on the Influence Mechanism of Internet Use on Rural Residents' Consumption LevelPLOS ONE

Dear Dr. wang,

Thank you for submitting your manuscript to PLOS ONE. After careful consideration, we feel that it has merit but does not fully meet PLOS ONE’s publication criteria as it currently stands. Therefore, we invite you to submit a revised version of the manuscript that addresses the points raised during the review process.

We look forward to receiving your revised manuscript.

Kind regards,

Kittisak Jermsittiparsert, Ph.D.

Academic Editor

PLOS ONE

Journal Requirements:

   "This is a part research accomplishment of the project“Research on the Evolution of Rural Residents' Consumption Behavior and the Coordinated Policy of Supply and Demand in the Past Forty Years of Reform" (No. 18BJL004))”, which is supported by Office of Chinese Philosophy and Social Sciences. "

6. Please amend either the title on the online submission form (via Edit Submission) or the title in the manuscript so that they are identical.

Additional Editor Comments:

According to reviewers, this article needs to be revised to a more appropriate quality. The authors strictly need to improve it based on their suggestions.

Reviewers' comments:

Reviewer's Responses to Questions

**Comments to the Author**

1. Is the manuscript technically sound, and do the data support the conclusions?

Reviewer #1: No

Reviewer #2: Yes

2. Has the statistical analysis been performed appropriately and rigorously? 

Reviewer #1: No

Reviewer #2: Yes

3. Have the authors made all data underlying the findings in their manuscript fully available?

Reviewer #1: No

Reviewer #2: Yes

4. Is the manuscript presented in an intelligible fashion and written in standard English?

Reviewer #1: No

Reviewer #2: Yes

5. Review Comments to the Author

Reviewer #1: Research work must have clear research problem, scope of the research is missing, similarly, methodology is also not clear, what about contribution author has not described contribution section. Contribution must in three fold first theoretical, that how it contributes to literature ? then methods and practically how it would be helpful ?

Reviewer #2: The paper is an interesting study, empirically studying the function mechanism of the internet use on the consumption level of rural residents and applying the instrumental variable method to solve the possible estimation bias caused by endogenous problems considering the survey data of 936 rural households in Jiangsu Province in China. This is an important work; well written and structured with only few literature review and comprehensive analyses. In general, this is an excellent contribution to knowledge in the respective field.

6. PLOS authors have the option to publish the peer review history of their article (what does this mean?). If published, this will include your full peer review and any attached files.

Reviewer #1: No

Reviewer #2: No

---

## [Author Response · Author response to Decision Letter 0]

20 Sep 2023

Dear reviewers,

Re: Manuscript ID: PONE-D-23-22564 & Title: Research on the Influence Mechanism of Internet Use on Rural Residents' Consumption Level

Thank you for your comments concerning our manuscript entitled “Research on the Influence Mechanism of Internet Use on Rural Residents' Consumption Level” (ID: PONE-D-23-22564). Those comments are all valuable and very helpful for revising and improving our paper, which has important guiding significance to our researches. We have studied comments carefully and made improvements accordingly, which we hope meet with your approval. All the revised parts are marked in red in the paper. The main corrections in the paper and the responses to the reviews are as follows: 

Responses to the comments of Reviewer #1: 

Many thanks to you for the comments to the point. The following are specific responses to each comment.

1. Is the manuscript technically sound, and do the data support the conclusions? No.

Response: The manuscript uses consumption literacy as a mediator variable and empirically tests the influence mechanism of internet use on the rural residents' consumption level in China through applying a mediation effect model, which is technically appropriate. The original plan of this study was to conduct a questionnaire survey in the eastern, central and western regions of China by using stratified random sampling method. However, as the COVID-19 made it difficult to conduct cross- provincial research, the research team had to adjust the plan and choose Jiangsu Province as the investigation area. The uneven level of economic development in Jiangsu can be seen as a microcosm of China's economic development, which can be divided into northern, central, and southern regions according to the degree of economic development from low to high. Therefore, the manuscript uses questionnaire survey data from Jiangsu Province to empirically study the influence mechanism of internet use on rural residents' consumption level, which has certain rationality, and the research conclusions obtained based on this also have certain credibility.

2. Has the statistical analysis been performed appropriately and rigorously? No.

Response: On the one hand, the manuscript conducts a descriptive statistical analysis of all variables involved in the econometric model in the section of "Model Setting and Variable Selection". On the other hand, the manuscript empirically studies the influence mechanism of internet use on the consumption level of rural residents in China by constructing a mediation effect model in the section of "Empirical Results and Analysis".

3. Have the authors made all data underlying the findings in their manuscript fully available? No.

Response: Due to paper length limitations, the manuscript did not provide raw data for the questionnaire survey, but the author can provide storage information of the data, which interested readers can download on their own.

4. Is the manuscript presented in an intelligible fashion and written in standard English? No.

Response: The manuscript is written in standard English, and the language of the manuscript is clear, correct and unambiguous.

5. Research work must have clear research problem. Scope of the research is missing, similarly, methodology is also not clear. What about contribution? Author has not described contribution section. 

Response: There is no consensus on the impact of internet use on residents' consumption level. Some academic researchers believe that internet use does not necessarily improve residents' consumption level in terms of total quantity. It only transfers traditional shopping channels to online, thus largely replacing offline consumption. However, other academic researchers believe that internet use not only reduces transaction costs in the consumption process and alleviates liquidity constraints, but also generates employment and income effects, thus significantly improving residents' consumption levels. The manuscript uses consumer literacy as a mediating variable and constructs a mediation effect model to study the influence of internet use on the consumption level of rural residents in China under the background of rapid development of new consumption driven by digital information technology.

6. Contribution must be in three fold: first theoretical (how it contributes to literature? ), then methods and practically how it would be helpful ?

Response: Under the circumstances that internet is naturally technology-driven and the increasing technology level of consumption products proposes higher requirements on consumers, some academic researchers began to study the effects and influences of residents’ consumption literacy when conducting research on the effects of internet on residents’ consumption level. However, most existing researches focus on theoretical analysis and lacks quantitative empirical research. The contributions of this manuscript mainly include the following aspects. (1) Based on the concept of consumer literacy, the research group designed 20 specific questions for measuring consumer literacy from five dimensions: consumer attitude, consumer motivation, consumer knowledge, consumer skills, and consumer traits. Drawing on the calculation method of financial literacy, the consumer literacy index was constructed to comprehensively evaluate the level of residents’ consumer literacy, thus enriching the theory of consumption economy. (2) Based on the survey data of rural households in Jiangsu Province, the manuscript empirically tested the influence mechanism of internet use on the rural residents' consumption level by constructing the theoretical analysis framework of “internet use - consumption literacy - rural residents consumption level” and taking consumption literacy as the mediator, thus expanding academic research on rural household consumption from a micro perspective. (3) By studying the influence mechanism of internet use on the consumption level of rural households and formulating relevant countermeasures and suggestions based on research conclusions, it not only helps to stimulate the consumption potential of rural residents, but also has extremely important practical significance and policy value for building a new development pattern centered on expanding the domestic circulation and achieving high-quality economic development in China.

Responses to the comments of Reviewer #2:

Special thanks to you for your favorable comments. The following is the specific response.

1.The paper is an interesting study, empirically studying the function mechanism of the internet use on the consumption level of rural residents and applying the instrumental variable method to solve the possible estimation bias caused by endogenous problems considering the survey data of 936 rural households in Jiangsu Province in China. This is an important work; well written and structured with only few literature review and comprehensive analyses. In general, this is an excellent contribution to knowledge in the respective field.

Response: According to the review comments, the author supplemented and improved the literature review section in the revised manuscript.

---

## [Decision Letter · Decision Letter 1]

7 Nov 2023

Research on the Influence Mechanism of Internet Use on Rural Residents' Consumption Level in China

PONE-D-23-22564R1

Dear Dr. wang,

We’re pleased to inform you that your manuscript has been judged scientifically suitable for publication and will be formally accepted for publication once it meets all outstanding technical requirements.

Kind regards,

Kittisak Jermsittiparsert, Ph.D.

Academic Editor

PLOS ONE

Additional Editor Comments (optional):

Reviewers' comments:

Reviewer's Responses to Questions

**Comments to the Author**

1. If the authors have adequately addressed your comments raised in a previous round of review and you feel that this manuscript is now acceptable for publication, you may indicate that here to bypass the “Comments to the Author” section, enter your conflict of interest statement in the “Confidential to Editor” section, and submit your "Accept" recommendation.

Reviewer #2: All comments have been addressed

Reviewer #3: All comments have been addressed

2. Is the manuscript technically sound, and do the data support the conclusions?

Reviewer #2: Yes

Reviewer #3: Yes

3. Has the statistical analysis been performed appropriately and rigorously? 

Reviewer #2: Yes

Reviewer #3: Yes

4. Have the authors made all data underlying the findings in their manuscript fully available?

Reviewer #2: Yes

Reviewer #3: Yes

5. Is the manuscript presented in an intelligible fashion and written in standard English?

Reviewer #2: Yes

Reviewer #3: Yes

6. Review Comments to the Author

Reviewer #2: The revisions completed accordingly with reviewers’ comments, which has now significantly improved the paper, presenting an innovative material on the subject, and therefore I would like to thank the authors for their contribution.

Reviewer #3: The research enjoys institutional support from the Office of Chinese Philosophy and Social Sciences, bolstering its credibility. However, it is vital to emphasize the omission of ethical review and approval in this study, as ethical oversight is fundamental in scientific research involving human subjects. The absence of ethical review raises ethical concerns that can impact the study's integrity and trustworthiness. Despite this, the research's innovative approach is evident in its title, "Exploring the Mechanisms of Internet Influence on the Consumption Levels of Rural Residents," which is underscored by a favorable plagiarism assessment. The study appropriately incorporates current literature and is methodologically sound, employing rigorous statistical analyses and robust data collection. However, while the two-stage least squares (2SLS) method is effectively used to address endogeneity, the study should also disclose the specific conditions and assumptions of Ordinary Least Squares (OLS) regression. The conclusions drawn are well-supported by the empirical results, demonstrating a positive correlation between internet use and rural residents' consumption levels, further validated by 2SLS estimation and mediation analysis. These findings substantiate the research hypotheses and provide valuable policy recommendations for enhancing rural residents' consumption levels within the context of internet development.

7. PLOS authors have the option to publish the peer review history of their article (what does this mean?). If published, this will include your full peer review and any attached files.

Reviewer #2: No

Reviewer #3: **Yes: **Srirath Gohwong, Ph.D.

---

## [Editor Report · Acceptance letter]

13 Nov 2023

PONE-D-23-22564R1 

Research on the Influence Mechanism of Internet Use on Rural Residents' Consumption Level in China——The Mediating Effect of Consumption Literacy 

Dear Dr. wang:

I'm pleased to inform you that your manuscript has been deemed suitable for publication in PLOS ONE. Congratulations! Your manuscript is now with our production department. 

Kind regards, 

on behalf of

Professor Kittisak Jermsittiparsert 

Academic Editor

PLOS ONE